# Rainbow peacock spiders inspire miniature super-iridescent optics

Bor-Kai Hsiung [1,8], Radwanul Hasan Siddique [2], Doekele G. Stavenga [3], Jürgen C. Otto[4], Michael C. Allen[5], Ying Liu[6], Yong-Feng Lu [6], Dimitri D. Deheyn [5], Matthew D. Shawkey [1,7] & Todd A. Blackledge[1]

Colour produced by wavelength-dependent light scattering is a key component of visual communication in nature and acts particularly strongly in visual signalling by structurally-coloured animals during courtship. Two miniature peacock spiders (*Maratus robinsoni* and *M. chrysomelas*) court females using tiny structured scales (~ 40 × 10 μm$^2$) that reflect the full visual spectrum. Using TEM and optical modelling, we show that the spiders' scales have 2D nanogratings on microscale 3D convex surfaces with at least twice the resolving power of a conventional 2D diffraction grating of the same period. Whereas the long optical path lengths required for light-dispersive components to resolve individual wavelengths constrain current spectrometers to bulky sizes, our nano-3D printed prototypes demonstrate that the design principle of the peacock spiders' scales could inspire novel, miniature light-dispersive components.

[1] Department of Biology and Integrated Bioscience Program, The University of Akron, Akron, OH 44325, USA. [2] Department of Medical Engineering, California Institute of Technology, Pasadena, CA 91125, USA. [3] Department of Computational Physics, University of Groningen, 9747 AG Groningen, The Netherlands. [4] 19 Grevillea Avenue, St. Ives, NSW 2075, Australia. [5] Scripps Institution of Oceanography (SIO), University of California, San Diego, La Jolla, CA 92093, USA. [6] Department of Electrical and Computer Engineering, University of Nebraska-Lincoln, Lincoln, NE 68588, USA. [7] Biology Department, Evolution and Optics of Nanostructures Group, Ghent University, Ledeganckstraat 35, 9000 Ghent, Belgium. [8] Present address: Scripps Institution of Oceanography (SIO), University of California, San Diego, La Jolla, CA 92093, USA. Correspondence and requests for materials should be addressed to B.-K.H. (email: bh63@zips.uakron.edu or (email: bkhsiung@ucsd.edu)

Controlling light through photonic micro- and nanostructures can transform human technology, including communications, sensing, security, and computing[1–3]. Biogenic photonic nanostructures have high translational potential[4] and reveal a diverse array of structural colour production mechanisms in plants and animals, including spiders[5–12]. In particular, some Australian peacock spiders can display extremely angle-dependent full-spectrum iridescence with high purity[13].

Iridescent integumentary colour patterns in plants and animals typically span only narrow segments of the visible wavelength range. For example, although its iridescent colour shifts from blue to violet, depending on the angle of light incidence, wings of *Morpho* butterflies are mostly blue[14,15]. The few organisms that exhibit a rainbow pattern, such as Bronzewing pigeons (*Phaps* spp.), do so using spatially segregated nanostructures along each feather, creating a gradient of colours from blue to red[16]. Thus, the colour of any particular point on the feather does not shift between discrete hues with change in viewing angle. Other structurally coloured feathers usually shift between only a few hues[17–19] and do not cover all colours in the visible spectrum. In contrast, the colour of abdominal scales from males of two miniature Australian peacock spiders, *Maratus robinsoni* (Supplementary Fig. 1a) and *M. chrysomelas* (Supplementary Fig. 1b) change from red to green to violet with slight movements (Fig. 1a, c, and https://youtube/eGS4JdewROU). These two species of peacock spiders (*Maratus* spp.) raise and wiggle their abdomens toward potential mates during courtship to display every colour across the entire visible spectrum (Supplementary Fig. 1), making this the first true rainbow-iridescent signal known in animals[13].

We hypothesize that the unique rainbow-iridescence in *M. robinsoni* and *M. chrysomelas* is produced by specialized abdominal scales that function as three-dimensional (3D) reflective diffraction grating structures. Two-dimensional (2D) diffraction grating-like structures are not as rare as previously thought, but are still uncommon in nature, and occur only in a handful of extant and fossil species[9,20–23], including plants[24–26]. Moreover, these previously described 2D diffraction gratings are likely epiphenomena that do not function in signalling, and are not then products of natural selection for optical functions[22,25]. Rainbow-iridescence is clearly a visual courtship signal in peacock spiders (Supplementary Fig. 1)[27]. Here, we first quantitatively characterize the colour and nanostructure of both peacock spider species using scanning and transmission electron microscopy (SEM/TEM), hyperspectral imaging (HSI), and imaging scatterometry. We then use analytical and finite-element optical simulation to identify the mechanism of colour production in the scales. Finally, we provide validation of the mechanism using bioinspired, actual-size physical prototypes via two-photon nanolithography. Our study identifies the mechanisms by which miniature Australian peacock spider actively display isolated wavelengths within visible spectrum during their courtship. This result is significant both because it is the first insight into this mechanism and because it could provide inspiration for the development of miniature light-dispersive components. By understanding biological design principles and emulating them through engineering, our research may allow light-dispersive components to perform under irradiances, and millimetre length scales, not possible before.

## Results

**3D airfoil-shaped grating**. *M. robinsoni* and *M. chrysomelas* have two types of visually distinct abdominal scales: rainbow-iridescent scales and velvet black scales (Fig. 2a, b and Supplementary Fig. 1). These scales show strikingly different morphologies: the

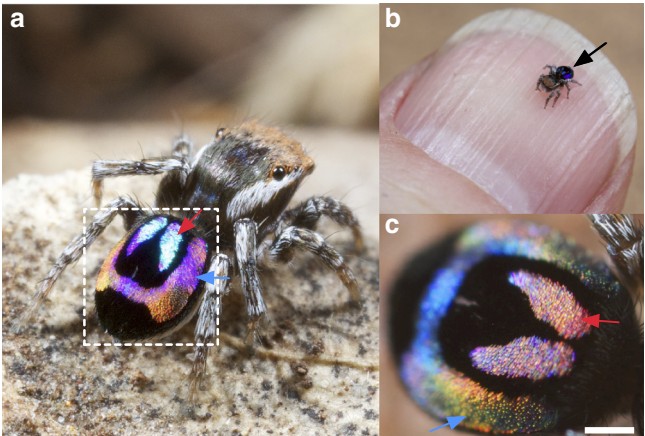

**Fig. 1** A miniature peacock spider with rainbow-iridescence. **a** An adult male *Maratus robinsoni*. **b** A *M. robinsoni* resting on a human fingernail: the spider is only ~ 2.5 mm in size. The iridescent abdomen of the spider is indicated by the black arrow. **c** A zoom-in view (scale bar: 200 μm) of the same spider abdomen as shown in the dashed square of **a**, but with different viewing angle. Note the colours of the iridescent patches almost change to their complementary colours between the two different views, from blue to red (red arrows), and from purple to yellow green (blue arrows)

black scales are brush-like and randomly oriented (Fig. 2c, d black arrowhead, Supplementary Fig. 2), while the rainbow-iridescent scales are more orderly aligned, cling to the cuticle surface and have bulky 3D shapes (Fig. 2c, d white arrowhead). Closer observation of the iridescent scales shows parallel grating structures on each individual scale for both spider species (Fig. 2e, f). The gratings are more regularly spaced on the scales of *M. robinsoni* (Fig. 2g) than those of *M. chrysomelas* (Fig. 2h). TEM on the transverse section of the iridescent scales reveals airfoil-shaped profiles (i.e. the curvatures are not concentric arcs; Fig. 2i, j). The surfaces of these airfoil-shaped scales are covered by prominent binary-phase grating structures with depths ~ 500 nm or more, and periods between 500 and 800 nm. Images of TEM sections (Fig. 2i, j) agree well with SEM images (Fig. 2e–h) and show the spacing of the gratings is more regular on the scales of *M. robinsoni* than that on the scales of *M. chrysomelas* (Table 1 and Supplementary Fig. 3). In addition, the gratings are asymmetrical between the two sides of the airfoil-shaped scales of *M. robinsoni*, with one side thinner and more densely distributed than the other, while the gratings are evenly, and more randomly, distributed on both sides of *M. chrysomelas* scales (Table 1).

**Separating the full visible spectrum over small angles**. The unique grating configuration of each *M. robinsoni* scale disperses the visible spectrum over a small angle, such that at short distances, the entire visible spectrum is resolved, and that a static microscopic rainbow pattern distinctly emerges (Fig. 3a). Hyperspectral analyses also demonstrate an array of full-spectrum reflectance spectra from the iridescent scales of both *M. robinsoni* (Fig. 3b) and *M. chrysomelas* (Supplementary Fig. 4a). On the basis of the SEM/TEM images, we hypothesize that the acute angle-sensitive rainbow-iridescence of *M. robinsoni* and *M. chrysomelas* result from the interaction of the surface nanograting and the microscopic airfoil-shape of the scales.

To evaluate our hypothesis, we design three different grating configurations, all using the same surface nanograting with a period of 670 nm (thickness: 170 nm, spacing: 500 nm), and a depth of 300 nm, but with different shapes/geometries: the first configuration is a conventional 2D grating (flat, Fig. 4a); the

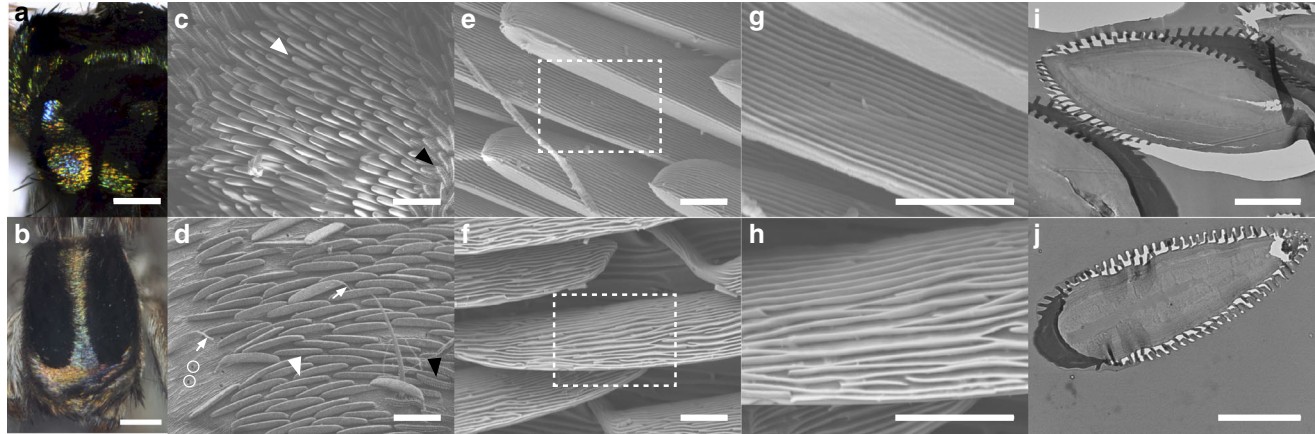

**Fig. 2** Optical and electron micrographs for the abdominal scales of *M. robinsoni* and *M. chrysomelas*. **a, b** Optical micrographs of the abdominal scales of *M. robinsoni* (**a**) and *M. chrysomelas* (**b**), showing two types of scales, iridescent and black. **c, d** SEM micrographs for the abdominal scales of *M. robinsoni* (**c**) and *M. chrysomelas* (**d**). At the centre of the image are the iridescent scales (white arrowhead). Black scales can be seen on both sides of the image (black arrowhead). The stems (white arrows) and the sockets (white circles) of detached scales can be observed. **e–h** Zoom-in views of SEM micrographs of the iridescent scales of *M. robinsoni* (**e, g**) and *M. chrysomelas* (**f, h**) showing grating structures. The grating period for the iridescent scales of *M. robinsoni* is more regular than that of *M. chrysomelas*. **i, j** TEM micrographs revealing the airfoil-shaped profiles and the surface nanogratings in the iridescent scales of *M. robinsoni* (**i**) and *M. chrysomelas* (**j**). Scale bar: **a, b** 200 µm; **c, d** 20 µm; **e–j** 5 µm

| Table 1 The thickness and spacing measured from TEM micrographs (Fig. 2i, j) | | | |
|---|---|---|---|
| | *M. robinsoni* | | *M. chrysomelas* |
| | Dense side (*n* = 18) | Sparse side (*n* = 16) | (*n* = 77) |
| Thickness (mean ± s.d.) | 167 ± 35 nm | 243 ± 35 nm | 203 ± 25 nm |
| Spacing (mean ± s.d.) | 306 ± 55 nm | 575 ± 47 nm | 381 ± 106 nm |

s.d. = standard deviation from the designated number of measurements (*n*)

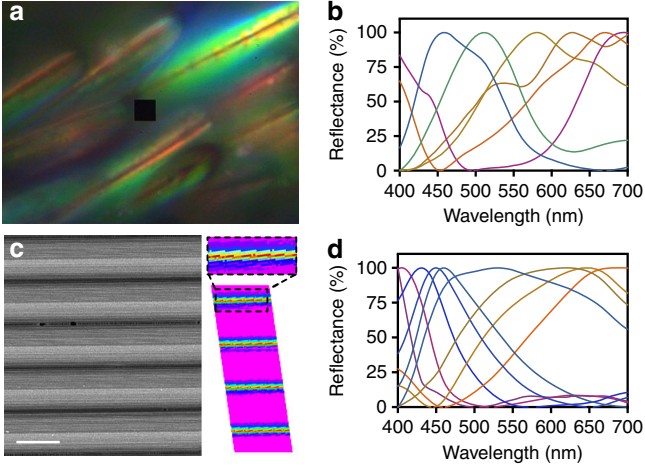

**Fig. 3** Observed microscopic colour patterns. **a** Light micrograph of rainbow patterned *M. robinsoni* scales. Black centre square: 4 × 4 µm². **b** Reflectance spectra collected by hyperspectral imaging of the iridescent scales. **c** SEM micrograph of the 3D printed foil grating (left, scale bar: 10 µm), and a hyperspectral image of the 3D printed foil grating (right) showing arbitrarily assigned, false-colour rainbow patterns emerging from the tip of the foil grating. Pixels show the same false-colour have the same reflectance spectrum and vice versa. **d** Reflectance spectra collected by hyperspectral imaging from the entire 3D printed foil grating image. For **b, d** the colours of the curves are estimated based on the "spec2rgb" function in R script "pavo"[49]

second configuration is a pentagonal prism that roughly resembles the shape of the scales with flat surfaces and abrupt joints (prism, Fig. 4b); lastly the third configuration is a spider scale-mimic structure, with nanogratings on the two lenticular curved (i.e., with concentric convex curvature) sides (foil, Fig. 4c). For the pentagonal prism configuration, the nanogratings are on the four upward-facing surfaces except for the base and the sides (Fig. 4b). This design is partly inspired by the iridescence-enhancing boomerang-shaped feather barbules of the bird-of-paradise *Parotia lawesii*[17,18].

We fabricate our designs using two-photon nanolithography, and verify the structure of the final products using SEM (Fig. 4e–g) and their optical output using hyperspectral imaging and scatterometry. These analyses demonstrate that only the foil grating (Fig. 3c, d) successfully reproduced the colour pattern from the spider scales (Fig. 3a, b), whereas the flat (Supplementary Fig. 4b) and prism gratings (Supplementary Fig. 4c) did not.

We further simulate the angle-dependent scattering spectra of the designed structures using finite-element modelling (FEM) and plot the simulated reflectance spectra of the three designed structures against their scattering/viewing angles (Fig. 5). The foil grating appears to show all the colours in a rainbow simultaneously in many simulated angles, but the rainbow pattern shows up especially well at three particular angles (Fig. 5a). This agrees well with the properties of the iridescent spider scales (Fig. 3a, b, Supplementary Fig. 4a) and the 3D printed foil grating (Fig. 3c, d). Nevertheless, the flat grating shows a pure colour at each

individual viewing angle, and can display most colours sequentially from 40° to 60° (Figs. 5b and 6c, g). By contrast, the prism grating cannot show either the rainbow pattern or the high purity colours (except for exactly 0°) (Fig. 5c). Again, these results match well with the results of hyperspectral analyses with the 3D printed flat (Supplementary Fig. 4b) and the prism (Supplementary Fig. 4c) gratings.

**Reversed diffraction order**. Imaging scatterometry further supports the detailed mechanism of the *M. robinsoni* and *M. chrysomelas* diffraction gratings (Fig. 6a, b). The order of the

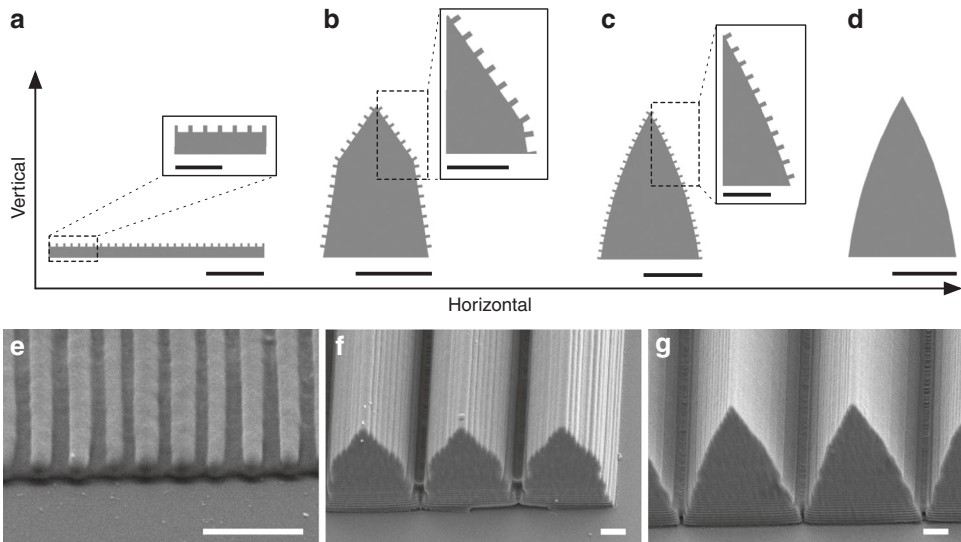

**Fig. 4** Different grating configuration designs and their SEM micrographs. **a** The flat grating. **b** The prism grating. **c** The foil grating. **d** The lenticular prism (foil-shaped structure without the surface nanograting). **e**–**g** The SEM micrographs for the flat (**e**), prism (**f**) and foil (**g**) gratings. Scale bar: **a**–**d** 5 μm; insets, 2 μm. **e**–**g** 2 μm

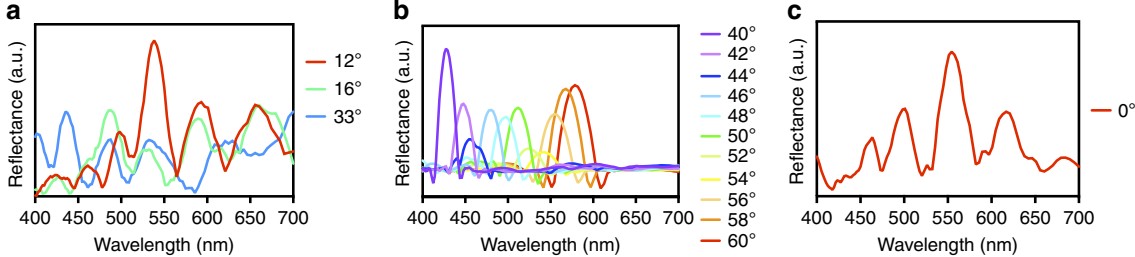

**Fig. 5** Simulated reflectance spectra. The simulated reflectance spectra plotted against different viewing angles based on results from Fig. 6g–i. Each spectrum is smoothed, normalized to the highest value of spectra plotted within the panel and assigned to an arbitrary colour. **a** The foil grating is capable of showing all the colours in a rainbow simultaneously in many different angles, and the rainbow pattern shows up particularly well at angles 12°, 16°, and 33°. **b** The flat grating can only show a single high purity colour at each viewing angle, and it takes > 20° rotation to shift the colour from one end of the spectrum to the other end (40°, peak wavelength ~ 400 nm → 60°, peak wavelength just below 600 nm). **c** The prism grating can only show the rainbow pattern at exactly 0°. In all other angles simulated, it cannot show the rainbow pattern like the foil grating (**a**), nor high purity colours as the flat grating (**b**) does

diffraction pattern of the spider scales is reversed relative to a conventional 2D (flat) grating (i.e., red⟶blue rather than blue⟶red, Fig. 6a, b vs. c). The scatterograms of the prism (Fig. 6d) and foil (Fig. 6e) gratings also show reverse-ordered diffraction patterns. This is because the surface gratings are oriented vertically on the airfoil-shaped scales, as previously reported in *Pierella* butterflies[28,29].

The FEM simulated angle-dependent scattering spectra of the designed structures are shown in Fig. 6f–i. To keep the simulated results in accordance with experimental results, we consider a plane-wave Gaussian pulse entering at the normal incident angle, and calculate the angle-dependent light scattering. The results of FEM optical simulation closely agree with the scatterograms showing reverse-ordered diffraction pattern for the prism (Fig. 6h), and foil (Fig. 6i) gratings.

The flat grating produces a discrete diffraction profile (Fig. 6g) allowing only three diffraction orders (−1, 0, 1) in the visible spectrum (400 ~ 700 nm). This is well predicted by the grating equation (Supplementary Note 1)[30]:

$$m_h \lambda_h = d_h(\sin\theta_i + \sin\theta_s) \qquad (1)$$

Here, $m_h$ is the diffraction order (or spectral order), which is an integer for a horizontal period of $d_h$, and $\theta_i$, $\theta_s$ are the incident and scattering (diffraction) angles, respectively. The reverse diffraction order can be explained by implementing the vertical grating equation and considering an exact vertical orientation of the surface grating to the surface normal[28,29]:

$$m_v \lambda_v = d_v(\cos\theta_i + \cos\theta_s) \qquad (2)$$

At normal incidence of light ($\theta_i$=0), the diffraction wavelength peak for a specific order varies proportionally to the cosine of the diffraction angle, which explains the curve-shaped reverse order diffraction profiles in Fig. 6h, i. To understand the microscopic shape effect, we further modified the vertical grating equation for a triangular horizontal grating. Considering the top angle of the triangular grating $\alpha$, the vertical grating equation is modified into:

$$m_v \lambda_v = 2d_v \cos(\alpha/2)(\cos\theta_i + \cos\theta_s) \qquad (3)$$

The superposition of Eq. 1 with Eq. 2 and Eq. 3 is plotted in Supplementary Fig. 5a and b, respectively. The higher order diffraction wavelength peak appears for a scattering angle $\theta_s = 0$

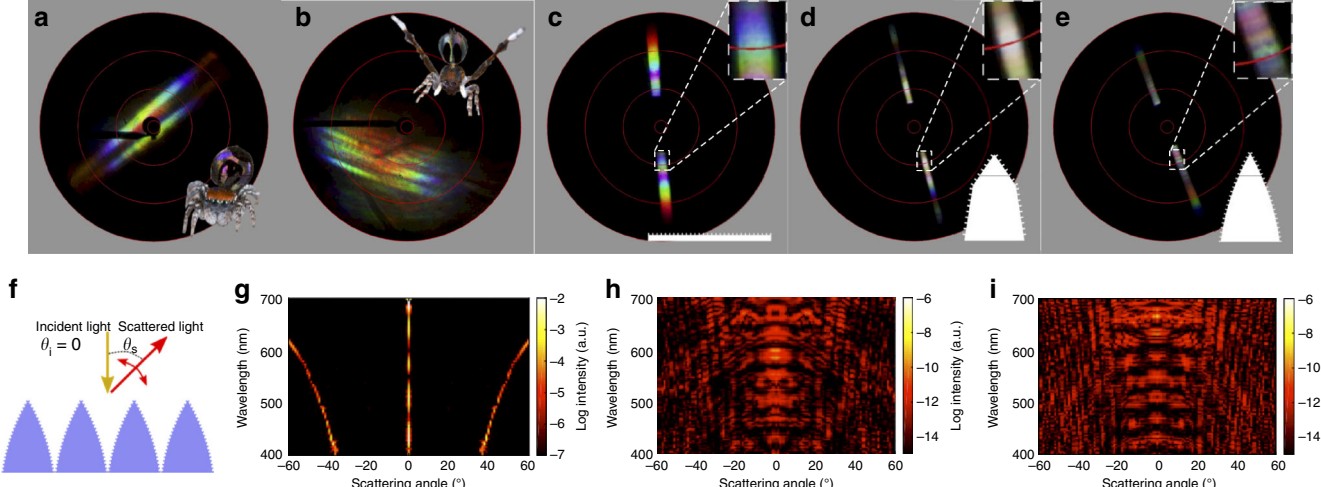

**Fig. 6** Imaging scatterograms and finite element optical simulation. **a–e** Scatterograms from iridescent scales of *M. robinsoni* (**a**), *M. chrysomelas* (**b**), from 3D printed flat grating (**c**), prism grating (**d**), and foil grating (**e**). The red circles from the centre out indicate 5°, 30°, 60°, 90° accordingly. The black centre of **c–e** is due to the 3D printed samples blocking the near-axis reflection in the scatterometer. Reverse-ordered diffraction patterns are observed in **a**, **b**, **d**, and **e** (see also insets). Fine banded patterns are observed in **d** and **e** only. **f**, Diagram of finite element optical simulations. **g–i** Results of finite element optical simulations for the 3D printed flat (**g**), prism (**h**) and foil (**i**) gratings. **h**, **i** Reverse-ordered diffraction patterns corresponding with **d** and **e**. Only **i** shows increased resolving power (i.e., finer diffraction features)

at normal incidence of light in the red spectral region due to the vertical surface grating period $d_v$ (Supplementary Fig. 5b). This explained the red colour of the reflection at the crest of the spider scales as well as the biomimetic (foil) prototype. Along the gradient of the scales, with increasing scattering angle, the colour changes from red to blue in an acute manner as seen in Fig. 3a. The abrupt microscopic shape of the prism grating might be the cause of the anomalous weak distribution of the diffraction pattern. However, the microscopic shape effect allows a large number of horizontal and vertical mode superposition in foil and prism gratings, thereby improving the diffraction efficiency (i.e., total diffracted power ($P$) over total incident power ($P_0$)).

**High resolving power**. The detailed fine features in both the experimental and simulated scattering profiles of the foil grating are clearly evident in Fig. 6e, i, respectively. Both prism and flat gratings show a coarser pattern in the scattering profiles than the foil grating (Fig. 6c, d, g, h), despite the fact that only the shape differs between the three types of gratings. These fine scattering features of the foil grating can be explained by its high resolving power (the ability to separate adjacent spectral lines of average wavelength $\lambda$) and low angular dispersion properties. The curved surfaces of the foil grating accumulate higher numbers of grooves under specific illumination conditions, in contrast to the flat grating. As the resolving power of a diffraction grating is proportional to the illuminated number of grooves and the periodicity[30], the microscopic shape provides an advantage for achieving high resolving power. To be precise, the microscopic triangular shape increases the number of grooves by a factor of cosec($\alpha/2$). That results in two times the number of effective grooves when $\alpha = 60°$ for a fixed illumination spot compared to a flat grating with the same period. Moreover, according to Eq. 3, the microscopic triangular shape reduces the angular dispersion (Note: not to be confused with chromatic dispersion, see Supplementary Note 2) for any order $m$ and period $d_v$ by a factor of 2cos($\alpha/2$). This reduces the angular spread of a spectrum of any order $m$. Therefore, the smaller angular spread of the foil grating enhances its "degree of iridescence" (here, we define the "degree of

iridescence" as the change in reflectance peak wavelength with the same amount of scattering angle variation) compared to regular binary phase gratings.

We further derived the vertical grating equation for the biomimetic foil grating from Eq. (3) by approximating the ellipsoidal curvature of the foil shape:

$$m_v \lambda_v = (\pi/\sqrt{2}) d_v \cos(\alpha/2)(\cos\theta_i + \cos\theta_s) \qquad (4)$$

According to Eq. 4, the curvature of the foil grating further increases the resolving power and decreases the angular spread by another ~ 10% ($\pi/\sqrt{8}$) when comparing the triangular grating with the same top angle (Eq. 3). Overall, the foil grating prototype is about twice as iridescent [($\pi/\sqrt{2}$)cos($\alpha/2$), $\alpha = 56°$, the top angle of the foil prototype] as a conventional 2D grating of the same period (flat). Hyperspectral analyses show that natural spider scales (Fig. 3b and Supplementary Fig. 4a) have an even higher resolving power than the foil prototype (Fig. 3d), suggesting that some aspect of the nanostructure (e.g. airfoil curvature) remains to be replicated and integrated into the next generation of prototypes to provide optimal resolving power closer to the natural system.

Due to the large period (~ 10 μm) of the microscopic grating, the angular separation between adjacent diffraction orders and the free spectral range of each individual order is small (Supplementary Fig. 6). However, introduction of the small-period (670 nm), surface vertical nanogratings modulates the horizontal diffraction orders and increases the diffraction efficiency and resolving power (Fig. 6i). This further demonstrates that resolving power increases due to the nanogratings on the microscopic curved surfaces, rather than simply angle and/or orientation[17,18] like that in the prism grating (Fig. 6h). The combination of vertical and horizontal grating effects in iridescent scales of *M. robinsoni* and *M. chrysomelas* provides saturated and intense diffraction outputs relative to the previously described natural example of a vertical grating in *Pierella* butterflies[28] (Supplementary Note 3). Due to the large horizontal period of *Pierella* butterfly scales (> 50 μm), the diffraction

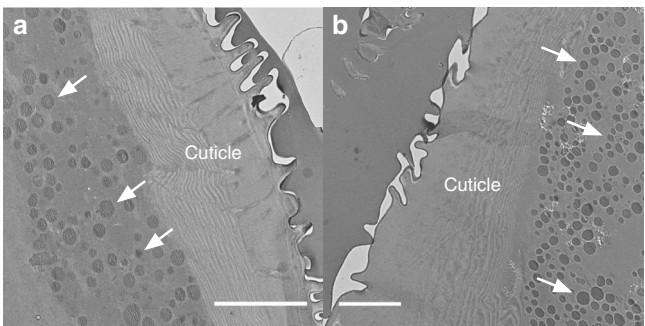

**Fig. 7** TEM micrographs of the abdominal cuticle. Abdominal TEM micrographs of *M. robinsoni* (**a**) and *M. chrysomelas* (**b**). The randomly organized dark granules in the hypodermis under the cuticle are melanosomes (white arrows). Scale bar: 5 μm

pattern is dominated by the vertical nanograting. We note that the banded pattern shown in the scatterograms of the prism (Fig. 6d) and foil (Fig. 6e) gratings is an artifact of the nanolithography production that results in the superposition of wavelengths (Supplementary Note 4). Since natural spider iridescent scales are partially disordered and not aligned in parallel (i.e., reduction in micrograting effect) (Fig. 2c, d) in the same manner as the synthetic ones (Fig. 4f, g), the banded pattern is not observed in the scatterograms of iridescent spider scales and the scattering pattern is mostly dominated by the vertical grating effect from the airfoil-shaped spider scales (Fig. 6a, b).

**Dark melanosome background**. We previously detected eumelanin in the black scales of *M. robinsoni* and *M. chrysomelas* using Raman spectroscopy[31]. TEM images of the *M. robinsoni* black scale sections show that eumelanin is diffusely and heterogeneously deposited in the black scales, because areas of different electron densities (shown in different levels of greyscale) and granular depositions were observed in the TEM micrographs of the scales (Supplementary Fig. 2b&c). Melanosomes are also observed in the hypodermis under the abdominal cuticle for both *M. robinsoni* (Fig. 7a) and *M. chrysomelas* (Fig. 7b), as previously reported in *M. splendens* and *M. anomalus*[32]. The dense layer of melanosomes likely functions to enhance the colour contrast.

## Discussion

We demonstrate here that the unique rainbow-iridescence of *M. robinsoni* and *M. chrysomelas* is produced by specialized 3D airfoil-shaped nanograting scales. These scales increase the resolving power of the diffraction grating through the synergistic effects of their vertically orientated surface nanogratings and the microscopic curvature of their airfoil-shape.

Contrast perceived by a visual system increases by accompanying black scales and an underlying black basal cuticle layer that decrease background scattering and increase the saturation of the structural colour, thus making the colour more conspicuous[33–35]. In addition, a regular grating period on the rainbow-iridescent scales of *M. robinsoni* further enhances iridescence (Fig. 1a). In contrast, the iridescence of *M. chrysomelas* (Supplementary Fig. 1b) is not as saturated due to its more irregular grating period[36]. In short, an innovative combination of regular surface nanogratings, airfoil-shaped microcurvatures, and background absorption mechanisms largely explain the striking iridescence observed in *M. robinsoni*.

It is unlikely that *M. robinsoni* and *M. chrysomelas* can discern the static microscopic rainbow patterns (Fig. 3a) of individual scales (Supplementary Note 5). However, the eyes of jumping spiders (Salticidae) have a high acuity (i.e., high spatial resolution)[37], and peacock spiders (*Maratus* spp.) likely possess tetrachromatic colour vision (i.e., high spectral resolution, personal communication with N.I. Morehouse; University of Cincinnati), so female peacock spiders are likely able to perceive the displayed colours in the form of dynamic rainbow iridescence. Male peacock spiders court females at a distance of a few centimetres, and raise and wiggle their abdomens to display the abdominal colours to the females (Supplementary Fig. 1). Therefore, the iridescent abdominal scales of *M. robinsoni* and *M. chrysomelas* produce the first known rainbow-iridescent signal in nature, and are likely a direct product of sexual selection through female choice. While functional hypotheses remain to be tested, the iridescence could enable mate recognition, provide females with honest information about male quality, or be the product of runaway selection[38].

Advanced computer-aided design allows construction of optical components that have a complex and previously unimaginable geometry/topology with novel functionality, high efficiency, and a compact footprint[39]. However, the solutions are local optimums that are largely confined by the initial input. On the other hand, nature provides unique and unexpected solutions for the design of advanced devices[40–43]. The design of high-efficiency light-extracting surfaces inspired by fireflies[44–46] show how these two seemingly different approaches act in synergy. Here, we identify the rainbow-iridescent scales of *M. robinsoni* and *M. chrysomelas* as a source of inspiration for designing miniature light-dispersive components with ultrahigh resolving power. Further improvement and optimization of miniature light-dispersive designs for specific applications can be made by incorporating computer-aided optical design processes. This powerful bioinspired approach would allow engineers to design and develop optical devices, especially spectrometers, with at least 50% smaller length scale (i.e., ~ an order of magnitude reduction in volume) for applications where fine-scale spectral resolution is required in a very small package, notably instruments on space missions, or wearable chemical detection systems. Therefore, a miniature spectrometer and light dispersive components will have significant impact to fields ranging from life sciences and biotechnology to material sciences and engineering.

## Methods

**Spider collection**. *M. robinsoni* specimens were collected on 22 October 2012 by Peter Robinson at the type locality near Newcastle, New South Wales (32° 59′ 50.42″ S, 151° 42′ 17.22″ E)[47]. *M. chrysomelas* (Simon, 1909) were collected by Jürgen Otto on October 2013 at several locations near Esperance in southwestern Western Australia[47]. *M. chrysomelas* is widespread in Australia, occurring in the east, west, central parts and the tropical north[47,48]. All specimens were preserved in 70% ethanol. Details of both species' distribution can be found in references[13,47].

**Light microscopy**. Specimens were observed using a 20/30 PV™ UV–visible-NIR microspectrophotometer (CRAIC Technologies, Inc.) with a 50× glass objective lens (numerical aperture (N.A.) = 0.7, free working distance = 1.1 mm, EC Epiplan 442060-9900-000, Carl Zeiss Microscopy, LLC.) for better brightfield colour imaging with extreme chromatic correction.

**Scanning electron microscopy**. To investigate the surface morphology of peacock spider scales, opisthosoma cuticle fragments were attached to sample stubs using carbon tape. Spider samples were sputter-coated with gold-palladium for 3 min under 20 mA, 1.4 kV and observed under a scanning electron microscope (JEOL 7401, Japan Electron Optics Laboratory Co. Ltd.) with 8 mm operating distance and 5 kV accelerating voltage.

**Transmission electron microscopy**. Opisthosoma cuticle fragments were dehydrated, and washed, followed by epoxy resin infiltration and embedding based on previously reported protocol(s)[8]. The cured epoxy block was trimmed with a Leica EM TRIM2 (Leica Microsystems) and microtomed into 80 nm thin sections using Leica EM UC6 (Leica Microsystems) with a DiATOME diamond knife (Electron Microscopy Sciences, Hatfield, PA, USA). Sections were mounted onto 100 mesh copper grids (EMS FCF100-Cu, Electron Microscopy Sciences) and observed under

a transmission electron microscope (JSM-1230, Japan Electron Optics Laboratory Co. Ltd., Akishima, Tokyo, Japan) without sample contrast staining for visualization.

**Two-photon nanolithography.** We used two-photon polymerization (TPP) nanolithography to fabricate the designed engineering prototypes (flat, prism and foil) for experimental investigations. The 3D laser lithography system (Photonic Professional GT, Nanoscribe GmbH, Germany) utilized a dip-in configuration with a 63×, 1.4 N.A. oil immersion objective lens (Zeiss, Germany) to focus the laser beam. An acrylic-based monomer liquid photoresist optimized for TPP applications ($n_r = 1.52$, IP-Dip, Nanoscribe GmbH) was drop-casted on a silicon wafer (500 μm thick with an oxidation layer of 3000 Å) and the objective lens immersed directly in the photoresist. A femtosecond laser (centre wavelength of 780 nm, pulse width of 100 fs, repetition rate of 80 MHz, and maximum power of 150 mW) was used as the irradiation source. A laser power of 25 mW was used in the TPP process and was controlled by an acousto-optic modulator. 50 mm/s writing speed was controlled by a galvo-mirror scanner[43]. Each design was fabricated to a 135 × 135 μm² area. The fabricated structures were then characterized using a Hitachi S-4700 SEM (Hitachi High-Technologies Corp., Tokyo, Japan), sputter-coated with 5 nm of chromium. The imaging voltage was kept low (<10 kV) to avoid damaging the structures.

**Hyperspectral Imaging.** A PARISS® hyperspectral imaging system (LightForm, Inc., Asheville, NC) was used to provide spatial mapping of the spectral output from the samples when normally illuminated with white light. Each sample was imaged without a coverslip for structure and spectral mapping. For structure 20% of the light output was imaged using a monochrome QIClick camera (QImaging); for spectral reflectance measurements, 80% of the light output was collected using a 100× 0.9 N.A. air objective (giving ≤ 0.5 μm spatial resolution) on a Nikon Eclipse 80i microscope with a PARISS® spectrometer utilizing a Retiga 2000DC CCD camera (QImaging). A 50 μm slit was used for window collection, and radiometric calibration was done with a Hg⁺/Ar⁺ lamp (LightForm Inc.), with spectral resolution measured better than 2 nm. A silver mirror (Thorlabs) was used as a reflectance reference for all measurements. Hyperspectral mapping was performed using a library of selected spectra with a minimum correlation coefficient (MCC) of 0.99 used as a discrimination factor to identify and map common spectra. All spectra were smoothed, normalized, and plotted using GraphPad Prism statistical software (GraphPad Software, Inc., La Jolla, CA, USA). The colours of the curves were estimated based on the "spec2rgb" function in R script "pavo"[49].

**Imaging scatterometry.** The spatial reflection characteristics of the iridescent spider scales and 3D printed grating designs were studied with an imaging scatterometer to visualize the diffraction patterns that they produced. The samples were glued to the tip of a glass micropipette, thus allowing us to position them in the first focal point of the ellipsoidal mirror of the imaging scatterometer[50]. Scatterograms were obtained by focusing a white light beam with narrow aperture (< 5°) onto at a small circular area (diameter ~ 13 μm) of the object, and the spatial distribution of the far-field scattered light was then monitored.

**Finite-element optical simulation.** The finite-element method was used to numerically simulate the variable-angle scattering profiles of the designed grating structures (flat, prism and foil). The Electromagnetic Waves, Frequency Domain interface of the commercial simulator COMSOL Multiphysics® was applied to individual geometry that consists of one unit cell of the gratings (Fig. 4a-d)[51]. Diffraction from the periodic micro- and nanostructures was simulated in two dimensions by utilizing Port conditions and S-parameters. The refractive index of the dielectric gratings was considered dispersion-less: $n_r = 1.52$. On either side of the unit cell, the Periodic Condition boundary condition with Floquet periodicity was used (Fig. 6f). This condition states that the solution on one side of the unit cell equals the solution on the other side multiplied by a complex-valued phase factor. The phase shift between the boundaries was evaluated from the perpendicular component of the wave vector.

The top and the bottom of the simulation unit cell were bounded by perfectly matched layers (PML). These boundary layers absorb any incident waves, preventing artifacts that could result from spurious interferences with re-entrant waves. Port boundary conditions were used to release the incident wave and to absorb the zeroth order (specular) reflected and transmitted waves. The input to each periodic port was an electric field amplitude vector with defined unit intensity. To assess angle-dependent scattering, the diffracting structures were rotated from −60° to 60° keeping the angle of light incidence at normal and zeroth order reflectance was calculated using S-parameters for wavelengths of 400–700 nm.

**Data availability.** All relevant data are available from the authors upon request.

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

## Acknowledgements

We thank David Hill for commenting on the manuscript and Nathan Morehouse for the peacock spider vision discussion before submission. This research was funded by the National Science Foundation (IOS-1257809, T.A.B.), Air Force Office of Scientific Research (FA9550-16-1-0331, M.D.S.; FA9550-15-1-0068, D.G.S.; FA9550-10-1-0555, D. D.D.), Human Frontier Science Program (RGY-0083, M.D.S.), Fonds Wetenschappelijk Onderzoek (G007177N, M.D.S.), The Scripps Institution of Oceanography Biomimicry for Emerging Science and Technology Initiative (D.D.D.) and The University of Akron Biomimicry Research and Innovation Center (B.-K.H.). B.-K.H. is supported by The Sherwin-Williams Company under a Biomimicry Fellowship.

## Author contributions

T.A.B., M.D.S., and B.-K.H. conceived research, and wrote the initial manuscript. B.-K.H. designed experiments, performed microscopy, and analysed data. R.H.S. performed optical modelling, simulations, and analysed data. D.G.S. performed imaging scatterometry. J.C.O. collected and documented spiders. M.C.A. and D.D.D. performed hyperspectral Imaging. Y.L. and Y.-F.L. performed two-photon nanolithography and prototype characterization. T.A.B. and M.D.S. provided scientific leadership to B.-K.H. All authors discussed the results and commented on the manuscript at all stages.

## Additional information

**Competing interests:** The authors declare no competing financial interests.

