## [Peer Review File · Nature Communications]

Reviewers' comments:

Reviewer #1 (Remarks to the Author):

The authors describe a novel photonic structure on spider abdomens, and do an admirably comprehensive job accounting for the observed visually striking and complex scattering using a combination of numerical modeling, analytical modeling, hyperspectral imaging, and then nanoscale 3D printing to validate their mechanistic explanation for the scattering they observe. They show that by wrapping a "standard" 2D diffraction grating around a scale with an air-foil-like shape and radius of curvature of ~ 100 's of microns, the resulting scattering is that of a broadband illuminant angularly separated into a full spectrum of saturated colors that are all observable at very small distances/angular displacements from the spider's body. This is distinct and novel relative to flat gratings or other curvatures, where large distances and/or angular separations are needed to observe the saturated colors resulting from the wavelength-dependent scattering of the whole broadband spectrum.

When viewed from the perspective of the methods section and figures alone, the results are pretty convincing. However, the rest of the manuscript as a whole suffers from some imprecise framing and imprecise writing that, as written, obscure the actual analytical work the authors have done.

It is true that subwavelength optics are interesting, complex, and important. It is also true that nature and the scientific literature are rife with an ever-increasing array of examples of well-described, sophisticated, subwavelength optics in organisms. So it seems to me that at this point in the evolution of this field, statements like "Light dispersion is crucial to fields ranging from life sciences and biotechnology to material sciences and engineering" are too general to be interesting or useful. What specific problem, or class of problems, might this "little rainbow" effect found in spiders be useful for? Really tiny spectrometers? If so, why are those useful? If there isn't (yet) an especially compelling, granular answer to these questions, then the paper's framing would be more convincing kept to the realm of basic science investigation, in the context of what is and isn't known about grating geometry and function (which is not a bad thing!).

I'm also unsure about the authors' use of the term "dispersion" here. I think I know what they mean, but most usually, this term describes how the refractive index of a material changes as a function of wavelength. Unless I'm missing something that should then be better explained in the paper, this isn't the effect they claim for the spider structures, but instead, the wavelength-dependent effects the authors document have to do with the near-wavelength geometry of the structure, and not with the relation between incident wavelength and refractive index or a material property per se. The spider effect seems to me something more like "wavelength-dependent scattering" or "complex

bidirectional reflectance distribution function" or just "diffraction", rather than "dispersion". If "dispersion" can be used in some contexts to describe any systematic wavelength effect regardless of the underlying cause, then this should be clear.

The feature that distinguishes the spider structures described here from simple flat gratings or other biological gratings is specifically the curvature of the scales on which grating-like ridges are found. For this reason, it was a little unsatisfying that this curvature is never quantified or further investigated, but only described as "not concentric arcs". What is special about the spider's curvature, and how can it be described, beyond just "not a concentric arc"? This seems to me to be the nub of their results, and is fairly readily quantifiable, but it isn't reported. If these "tiny rainbows"/high resolution diffraction effect is as important as they say, then it seems as important to then quantify the curvature that gives rise to them. Also, it gets a little confusing which aspect is "horizontal" and which is "vertical" - an additional schematic would be helpful in that respect.

The authors also write that this shape enhances the "degree of iridescence" compared to other gratings they considered, and later in the discussion say that the work they've done "explains the striking iridescence". "Iridescence" isn't a term with much particular physical meaning, so I wasn't sure what specifically they were trying to claim in context. My advice would be to avoid this term altogether, in favor of specific physical statements about the effect of interest in each case.

I'm also not sure it is fair to claim that they have found "the first rainbow-iridescent signal" in nature, given that they conclude that spiders are unlikely to be perceiving the full reflected, spatially separated spectrum at any given time. In order to be a signal, a stimulus needs to be received, but the authors argue that the tiny rainbow per se probably isn't received in this case since the spiders' angular acuity is likely too low. The more interesting question to me is, given the comparatively low angular resolution of spiders' eyes compared to this rainbow, and the very complex scattering effects from the scales at relevant lengthscales, what then is the salient part of this signal to the spider? Would that give any more clues as to what the most physically interesting features are likely to be? What would these diffraction patterns look like to something with many eyes, low spatial acuity, but high spectral resolution (as I think I understand the spiders to be)? Without considering this issue in more experimental detail, it would be my advice to avoid making any "the first" claims, and just focus on what is especially interesting and demonstrably true about the structure.

Minor comments: extended figure 5: "scar bar"; some editing mistakes around line 585;

Reviewer #2 (Remarks to the Author):

I have reviewed this submission to Nature communications with interest, but frankly I have to confess that the more I read the more disappointed I got on the document. Perhaps I was moved initially by the title and abstract to expect something extraordinary, but this is truly not the case.

I find their claims of extraordinary optical properties, really lacking support. The diffraction presented by this spiders did not strike to me as anything remarkable, it is just a nanostructured mounted on a microstructure. Very much a like the one presented by butterflies, but clearly with its on particularities. The diffraction is not selective, quite broadband actually.

their central claim "scales achieve resolving power beyond the performance of conventional 2D diffraction gratings" seems unfair to diffraction gratings. I am quite positive one can give the required performance to an optical engineer and most likely a solution will be found with standard technology. After all, what we see here is a diffractive structure just riding a non-flat microstructure.

I give actually a bit more credit to the group that simulated and fabricated the artificial replica via 2 photon lithography. That seems nice but it is not a technological feat.

So, there is nothing particularly wrong here, in fact they present a substantial amount of well done work, but in my opinion this paper's impact is modest. It is already a cliché to look in nature for inspiration, but in this particular instance the structure is not even hard to identify or reproduce. So, to suggest it might change how optical designers think or imagine building dispersive structure is quite an exaggerated view.

My suggestion is pick a more specialized journal.

Reviewer #3 (Remarks to the Author):

This paper reports a very thorough study of an original and unusual case of iridescence in nature, and employs the best possible methods to determine the structures involved and the precise optical

reflections (I particularly like the scatterometer), and to characterize the optical effect. The level of effort and care to gather data from such small scales pays off when the authors can reveal the important effect of a 3D (rather than flat) surface which houses the diffraction gratings. The engineered devices are useful to confirm the principles hypothesised (as always with such studies, the thoughts on commercial applications require specialized and extensive study). I believe that such a comprehensive study has led to results that be trusted and therefore this is a valuable contribution that will interest researchers in many fields. I recommend that this paper is published after minor editing.

Andrew Parker

Point-by-point responses to the reviewers' comments on the manuscript

"Rainbow peacock spiders inspire miniature super iridescent optics"

We thank the reviewers for their careful reading of our manuscript and for their comments and suggestions to improve the quality of the text. The following responses address all of the reviewers' comments in a point-by-point fashion.

Reviewer #1

Specific Comments

Comment #1 *"The authors describe a novel photonic structure on spider abdomens, and do an admirably comprehensive job accounting for the observed visually striking and complex scattering using a combination of numerical modeling, analytical modeling, hyperspectral imaging, and then nanoscale 3D printing to validate their mechanistic explanation for the scattering they observe. They show that by wrapping a "standard" 2D diffraction grating around a scale with an air-foil-like shape and radius of curvature of ~100's of microns, the resulting scattering is that of a broadband illuminant angularly separated into a full spectrum of saturated colors that are all observable at very small distances/angular displacements from the spider's body. This is distinct and novel relative to flat gratings or other curvatures, where large distances and/or angular separations are needed to observe the saturated colors resulting from the wavelength-dependent scattering of the whole broadband spectrum.*

When viewed from the perspective of the methods section and figures alone, the results are pretty convincing. However, the rest of the manuscript as a whole suffers from some imprecise framing and imprecise writing that, as written, obscure the actual analytical work

the authors have done.

It is true that subwavelength optics are interesting, complex, and important. It is also true that nature and the scientific literature are rife with an ever-increasing array of examples of well-described, sophisticated, subwavelength optics in organisms. So it seems to me that at this point in the evolution of this field, statements like "Light dispersion is crucial to fields ranging from life sciences and biotechnology to material sciences and engineering" are too general to be interesting or useful. What specific problem, or class of problems, might this "little rainbow" effect found in spiders be useful for? Really tiny spectrometers? If so, why are those useful? If there isn't (yet) an especially compelling, granular answer to these questions, then the paper's framing would be more convincing kept to the realm of basic science investigation, in the context of what is and isn't known about grating geometry and function (which is not a bad thing!)."

Response: We thank the reviewer for the great summary of our research and opinion about the framing of this manuscript. We modified the manuscript and reframed it as potential biological inspiration for future designs for miniature light-dispersive components. We have also explained why and how these miniature designs could have a large impact in fields from life science and biotechnology to material sciences and engineering in the Discussion, for example, small and powerful spectrometers that could be contained within wearable devices could help soldiers and explorers avoid hazardous environments in war zones or during expeditions. But this is only one example, and more extensive information is provided in Line 298~305.

Comment #2 *"I'm also unsure about the authors' use of the term "dispersion" here. I think I know what they mean, but most usually, this term describes how the refractive index of a material changes as a function of wavelength. Unless I'm missing something that should then be better explained in the paper, this isn't the effect they claim for the spider structures, but instead, the wavelength-dependent effects the authors*

document have to do with the near-wavelength geometry of the structure, and not with the relation between incident wavelength and refractive index or a material property per se. The spider effect seems to me something more like "wavelength-dependent scattering" or "complex bidirectional reflectance distribution function" or just "diffraction", rather than "dispersion". If "dispersion" can be used in some contexts to describe any systematic wavelength effect regardless of the underlying cause, then this should be clear."

Response: We thank the reviewer for pointing this out. Indeed, the term “Dispersion” can be used to describe any systematic wavelength effect regardless of the underlying cause. Therefore, “Dispersion” under the context of a diffraction grating will have a different definition than that of a prism. We made this clear to the readers by adding a new paragraph in Supplementary Note 2.

Comment #3 *“The feature that distinguishes the spider structures described here from simple flat gratings or other biological gratings is specifically the curvature of the scales on which grating-like ridges are found. For this reason, it was a little unsatisfying that this curvature is never quantified or further investigated, but only described as "not concentric arcs". What is special about the spider's curvature, and how can it be described, beyond just "not a concentric arc"? This seems to me to be the nub of their results, and is fairly readily quantifiable, but it isn't reported. If these "tiny rainbows"/high resolution diffraction effect is as important as they say, then it seems as important to then quantify the curvature that gives rise to them. Also, it gets a little confusing which aspect is "horizontal" and which is "vertical" - an additional schematic would be helpful in that respect.”*

Response: We agree with the reviewer’s suggestion. In fact, we indeed attempted to quantify it. However, as the curvature of the natural spider scales does not follow any spherical /circular shape (i.e. freeform curvatures), it was not straightforward to define the radius of curvature in a quantitative manner. We have already

analytically shown that the microscopic triangular shape has significant impact on the grating performance.

Nevertheless, following the reviewer's suggestion, we derived Equation 4 considering ellipsoidal curvature. Several previous literatures (for example, H. Noda, T. Namioka, and M. Seya, "Geometric theory of the grating," J. Opt. Soc. Am. 64, 1031-1036 (1974)) implied that curvature effect modifies the effective grating periodicity. According to the newly derived Eq. 4, we found that the effective grating period changes with a factor of $\pi/\sqrt{8}$ and indeed, the curvature effect improves grating performance roughly 10% in addition to the macroscopic shape. This is now added in the manuscript in Line 207~213. We also modified Fig. 4 to make it clear what do we mean by "horizontal" and "vertical".

Comment #4 *"The authors also write that this shape enhances the "degree of iridescence" compared to other gratings they considered, and later in the discussion say that the work they've done "explains the striking iridescence". "Iridescence" isn't a term with much particular physical meaning, so I wasn't sure what specifically they were trying to claim in context. My advice would be to avoid this term altogether, in favor of specific physical statements about the effect of interest in each case."*

Response: Iridescence is usually defined as a "change in hue of a surface with varying observation angles" (doi:10.1126/science.1173324). Hence, in this manuscript we define the "degree of iridescence" as "the change in hue with the same amount of scattering angle variation", and use this definition as the basis for our quantification. The definition has been clarified in our manuscript (Line 204~206).

Comment #5 *"I'm also not sure it is fair to claim that they have found "the first rainbow-iridescent signal" in nature, given that they conclude that spiders are unlikely to be perceiving the full reflected, spatially separated spectrum at any given time. In order to be a signal, a stimulus needs to be received, but the authors argue that the tiny rainbow per se probably isn't received in this case since the spiders' angular acuity is likely too low. The more interesting question to me is,*

given the comparatively low angular resolution of spiders' eyes compared to this rainbow, and the very complex scattering effects from the scales at relevant length scales, what then is the salient part of this signal to the spider? Would that give any more clues as to what the most physically interesting features are likely to be? What would these diffraction patterns look like to something with many eyes, low spatial acuity, but high spectral resolution (as I think I understand the spiders to be)?

Without considering this issue in more experimental detail, it would be my advice to avoid making any "the first" claims, and just focus on what is especially interesting and demonstrably true about the structure."

Response: We agree with the reviewer that for anything to be a “signal”, it has to be perceivable by the intended receivers. With that said, we argue that this is indeed “the first rainbow-iridescent signal” in nature. As iridescence is defined as “change in hue over varying observation angles”, the essence of an “iridescent signal” is that it is “dynamic” (doi:10.1126/science.1173324, doi:10.1016/j.cobeha.2015.08.007). Therefore, while female spiders can probably not perceive the “static” rainbow, their exceptional spectral resolution (tetrachromacy) makes it likely that they can perceive the change in hue from individual scales. Females have the acuity and spectral resolution to perceive colour variation across the male’s abdomen. This emphasizes our point that the iridescence itself is likely the salient portion of the visual signal, and we have added some text to the discussion on this point. See our explanation in the text at Line 269~281.

Comment #6 “*extended figure 5: "scar bar"; some editing mistakes around line 585*”

Response: We fixed the typo and grammar. Thank you.

Reviewer #2

General Comments

Comment #1: *“I have reviewed this submission to Nature communications with interest, but frankly I have to confess that the more I read the more disappointed I got on the document. Perhaps I was moved initially by the title and abstract to expect something extraordinary, but this is truly not the case.”*

Response: We are glad to hear that the title and abstract of this manuscript gathered the reviewer’s attention and interests. The changes made to the manuscript substantially increase its novelty and impact as detailed below. Critical changes can be found in Line 30~41, Line 83~89, Line 187~220, Line 269~281, and Line 295~305. These changes address the broad sense of the document and provide a stronger story to the observations.

Comment #2: *“I find their claims of extraordinary optical properties, really lacking support. The diffraction presented by this spiders did not strike to me as anything remarkable, it is just a nanostructured mounted on a microstructure. Very much a like the one presented by butterflies, but clearly with its on particularities. The diffraction is not selective, quite broadband actually.*

their central claim "scales achieve resolving power beyond the performance of conventional 2D diffraction gratings" seems unfair to diffraction gratings. I am quite positive one can give the required performance to an optical engineer and most likely a solution will be found with standard technology. After all, what we see here is a diffractive structure just riding a non-flat microstructure.

I give actually a bit more credit to the group that simulated and fabricated the artificial replica via 2 photon lithography. That seems nice but it is not a technological feat.

So, there is nothing particularly wrong here, in fact they present a substantial amount of well done work, but in my opinion this paper's

impact is modest. It is already a cliché to look in nature for inspiration, but in this particular instance the structure is not even hard to identify or reproduce. So, to suggest it might change how optical designers think or imagine building dispersive structure is quite an exaggerated view.”

Response: The extraordinary optical properties of these spider scales and how they are better than conventional 2D gratings are shown in qualitative ways in revised Figure 5, Supplementary Figure 4&8. We now add quantitative analyses to show the biomimetic Foil grating is about twice as iridescence as the conventional 2D grating with the same period (the Flat grating). We are happy to hear that the reviewer confirmed our research was substantially well done and correct. And we agree that biomimicry per se is not novel, but definitely not mainstream yet and still an emerging field – its utility is evidenced by its increasingly common use. However, the example we present in this research is powerful in that it achieves two-fold better performance than conventional technology. The innovation (spider scale-inspired 3D grating structure) may seem “straightforward” in design but it had not been applied before our study. The peacock spider clearly was a key inspiration for this new technology. Moreover, no one has ever investigated the optical outputs resulting from the interactions between nanoscale grating structures and microscale complex 3D geometries before. Therefore, our research may open a door to new design strategies for the optical engineers to explore.

Comment #3: *“My suggestion is pick a more specialized journal.”*

Response: We thank the reviewer’s opinion, we are afraid that we disagree. We think *Nature Communications* is the best publishing avenue for this research due to the interdisciplinary nature of the research, and its potential applications and impacts. Thank you.

Reviewer #3

General Comments

Comment #1 *“This paper reports a very thorough study of an original and unusual case of iridescence in nature, and employs the best possible methods to determine the structures involved and the precise optical reflections (I particularly like the scatterometer), and to characterize the optical effect. The level of effort and care to gather data from such small scales pays off when the authors can reveal the important effect of a 3D (rather than flat) surface which houses the diffraction gratings. The engineered devices are useful to confirm the principles hypothesised (as always with such studies, the thoughts on commercial applications require specialized and extensive study). I believe that such a comprehensive study has led to results that be trusted and therefore this is a valuable contribution that will interest researchers in many fields. I recommend that this paper is published after minor editing.*
Andrew Parker”

Response: We thank the reviewer for these positive comments. We agree that this research will generate a lot of interests and impact from many fields due to its inherent interdisciplinary scope, including but not limiting to, photonic engineers, physicists, and biologists. Thank you!

Reviewers' comments:

Reviewer #1 (Remarks to the Author):

The manuscript is much improved. It is more readable, and it is easier to understand the technical arguments for the novelty of the structure described here. While I followed the overall written argument much more readily this time, the writing still suffers from using terms that don't have a technical meaning to describe specific, geometric scattering phenomena. I'd argue that in this context, where you are trying to communicate to a diverse scientific audience, and you'd like for engineers to be able to understand what this work might have to offer their applications, it is critically important to use language specific to the exact scattering effect you are describing. If you genuinely want people to make spectrometers based on this work, the scientific language needs to be precise. Rather than "pitching" the coolness of this structure to the audience, it would be better and ultimately more convincing to describe it very carefully and let the audience decide for themselves if it is cool/useful.

Please see below for many places in the manuscript that would benefit from more precise language:

line 73, this is a suggested rewrite, it is a little ambiguous as written: these previously described 2D diffraction gratings are likely epiphenomena that do not function in signaling, and are not then products of natural selection for optical function.

Line 84: "actively display all visible colours"

I think what you mean is "isolate, in order, all visible wavelengths in space"

"Colour" is a construct of environmental radiance interacting with eyes/brains...

Line 89: by "illumination conditions and at small distances" is more precisely "irradiances" and "millimeter length scales"

line 113: Section heading: "Dispersing light over a short distance", more specifically the relevant metric is "Separating the full spectrum of visible light over small angles"

115: "resolves broadband light over a very short range and angle." More specifically this is "disperses the visible spectrum over a small angle, such that at short distances, the entire visible spectrum is resolved"

142: "exceptional nature". Exception from what? Maybe "detailed mechanism" is more appropriate?

Line 184: "diffraction efficiency" is not defined. What's efficient here? The amount of light scattered? The separation of individual wavelengths?

line 189: What does "coarser" scattering mean?

Line 205: "Hue" is also a complex property of color, which is then a perceptual construct. I think this means change in maximum wavelength of the reflected spectrum with angular position.

213: "superior optical properties" - superior for what context? Optimization depends on the task at hand. Be more specific about what property is different in the spider grating vs. the other structures tested.

214: "twice as iridescent" still obscures more than it explains. Just say that you get twice the change in maximum reflected wavelength for a given solid angle. (I think that is what you meant, but if I still don't understand, it is because "twice as iridescent" is still pretty vague).

230: "Rich and strong diffraction outputs". "Rich and strong" don't have an optical meaning. More specific words would be "saturated and intense", if this is what you meant.

Supplementary Note 2 is really helpful - can a shortened version of this information go in the main MS?

Also, "resolving power" is not specifically defined - since it comes up a lot, it would also help to define this for a general audience.

Line 260: "Iridescence is enhanced". I think in this context you mean "contrast perceived by a visual system increases"

In the conclusion, it is helpful to have some specific suggestions about applications in the conclusion but this now reads as a little over-specific. Is it possible to be intermediately general? What about: "will reduce spectrometer volumes by an order of magnitude, for applications where fine-scale spectral resolution is required in a very small footprint, notably instruments on space missions, or wearable chemical detection systems". Spectrometers an order of magnitude volume smaller, and therefore wearable or more space-worthy strikes me as something genuinely interesting and grounded in the scientific claims of the paper.

Point-by-point responses to the reviewers' comments on the manuscript

"Rainbow peacock spiders inspire miniature super iridescent optics"

Reviewer #1

The manuscript is much improved. It is more readable, and it is easier to understand the technical arguments for the novelty of the structure described here. While I followed the overall written argument much more readily this time, the writing still suffers from using terms that don't have a technical meaning to describe specific, geometric scattering phenomena. I'd argue that in this context, where you are trying to communicate to a diverse scientific audience, and you'd like for engineers to be able to understand what this work might have to offer their applications, it is critically important to use language specific to the exact scattering effect you are describing. If you genuinely want people to make spectrometers based on this work, the scientific language needs to be precise. Rather than "pitching" the coolness of this structure to the audience, it would be better and ultimately more convincing to describe it very carefully and let the audience decide for themselves if it is cool/useful.

Authors' response: We thank the reviewer for the invaluable suggestions continuously helping us to improve our manuscript.

Please see below for many places in the manuscript that would benefit from more precise language:

line 73, this is a suggested rewrite, it is a little ambiguous as written: these previously described 2D diffraction gratings are likely epiphenomena that do not function in signaling, and are not then products of natural selection for optical function.

Authors' response: Revised according to suggestion (line 74~76).

Line 84: "actively display all visible colours"

I think what you mean is "isolate, in order, all visible wavelengths in space"

"Colour" is a construct of environmental radiance interacting with eyes/brains...

Authors' response: replaced "all visible colours" with "isolated wavelengths within visible spectrum" (line 87).

Line 89: by "illumination conditions and at small distances" is more precisely "irradiances" and "millimeter length scales"

Authors' response: Revised according to suggestion (line 92~93).

line 113: Section heading: "Dispersing light over a short distance", more specifically the relevant metric is "Separating the full spectrum of visible light over small angles"

Authors' response: Changed the section heading to "Separating full visible spectrum over small angles" (line 117).

115: "resolves broadband light over a very short range and angle." More specifically this is "disperses the visible spectrum over a small angle, such that at short distances, the entire visible spectrum is resolved"

Authors' response: Revised according to suggestion (line 118~120).

142: "exceptional nature". Exception from what? Maybe "detailed mechanism" is more appropriate?

Authors' response: Revised according to suggestion (line 146).

Line 184: "diffraction efficiency" is not defined. What's efficient here? The amount of light scattered? The separation of individual wavelengths?

Authors' response: diffraction efficiency is now defined as "total diffracted power (P) over total incident power (P_0)" (line 189).

line 189: What does "coarser" scattering mean?

Authors' response: It now reads as "coarser pattern in the scattering profiles" (line 193).

Line 205: "Hue" is also a complex property of color, which is then a perceptual construct. I think this means change in maximum wavelength of the reflected spectrum with angular position.

Authors' response: Although "Hue" (unlike its colloquial usage) is a technical terminology among biologists who study colors, and is defined exactly as the "wavelength of peak reflectance" (Montgomerie 2006 (chapter: analyzing colors in Bird Coloration Vol. 1 ISBN: 0674018931) & Maia 2013 (DOI: 10.1111/2041-210X.12069)), we spelled it out as "maximum reflected wavelength" (line 211) in this manuscript to avoid confusion that may arise for readers from other fields.

213: "superior optical properties" - superior for what context? Optimization depends on the task at hand. Be more specific about what property is different in the spider grating vs. the other structures tested.

Authors' response: We deleted this clause, since it is already compared quantitatively and very specifically in the sentence preceding this clause.

214: "twice as iridescent" still obscures more than it explains. Just say that you get twice the change in maximum reflected wavelength for a given solid angle. (I think that is what you meant, but if I still don't understand, it is because "twice as iridescent" is still pretty vague).

Authors' response: The reviewer's understanding is correct. However, we would like to keep it as is here. "Iridescent/iridescence" is a well established trait in biological literature (DOI: 10.1098/rsif.2008.0395.focus, DOI: 10.1098/rsif.2009.0013.focus, DOI: 10.1098/rsif.2008.0354.focus, DOI: 10.1111/nph.13066). Since we already defined how we quantify "iridescence"

in Line 210~212, we argue that keeping “twice as iridescence” here will not obscure the understanding to readers from other fields, and can better disseminate the idea to biologists.

230: "Rich and strong diffraction outputs". "Rich and strong" don't have an optical meaning. More specific words would be "saturated and intense", if this is what you meant.

Authors' response: Revised according to suggestion (line 235).

Supplementary Note 2 is really helpful - can a shortened version of this information go in the main MS?

Authors' response: The following note was added into the main text – “Note: not to be confused with chromatic dispersion, see Supplementary Note 2” (line 206~207).

Also, "resolving power" is not specifically defined - since it comes up a lot, it would also help to define this for a general audience.

Authors' response: Resolving power is now defined as “the ability to separate adjacent spectral lines of average wavelength λ ” (line 196~197).

Line 260: "Iridescence is enhanced". I think in this context you mean "contrast perceived by a visual system increases"

Authors' response: Revised according to suggestion (line 266).

In the conclusion, it is helpful to have some specific suggestions about applications in the conclusion but this now reads as a little over-specific. Is it possible to be intermediately general? What about: "will reduce spectrometer volumes by an order of magnitude, for applications where fine-scale spectral resolution is required in a very small footprint, notably instruments on space missions, or wearable chemical detection systems". Spectrometers an order of magnitude volume smaller, and therefore wearable or more space-worthy strikes me as something genuinely interesting and grounded in the scientific claims of the paper.

Authors' response: Revised according to suggestion (line 305~307).